# Effects of Aging on New Bone Regeneration in a Mandibular Bone Defect in a Rat Model

**DOI:** 10.3390/biomimetics9080466

**Published:** 2024-08-01

**Authors:** Jung Ho Park, Jong Hoon Park, Hwa Young Yu, Hyun Seok

**Affiliations:** 1Department of Orl and Maxillofacial Surgery, School of Dentistry, Jeonbuk National University, Jeonju 54896, Republic of Korea; junghopark1106@gmail.com (J.H.P.); themobius8@naver.com (J.H.P.); 2Department of Oral Pathology, School of Dentistry, Jeonbuk National University, Jeonju 54896, Republic of Korea; enzks17@gmail.com; 3Research Institute of Clinical Medicine of Jeonbuk National University-Biomedical Research Institute of Jeonbuk National University Hospital, Jeonju 54907, Republic of Korea

**Keywords:** aging, new bone regeneration, maxillofacial bone, mandible

## Abstract

The effects of aging on the healing capacity of maxillofacial bone defects have not been studied. The aim of this study was to evaluate the effects of aging on the regeneration of round bony defects in the mandible. We created a round-shaped bony defect in the mandibular angle area in rats of different ages (2-[2 M], 10-[10 M], and 20-month-old [20 M]) and evaluated new bone regeneration in these groups. Changes in bone turnover markers such as alkaline phosphatase, procollagen type I N-terminal propeptide (PINP), cross-linked C-telopeptide of type I collagen, and tartrate-resistant acid phosphatase 5B (TRAP5b) were investigated. The bone volume/total volume and bone mineral density of the 20 M group were significantly higher than those of the 2 M group (*p* = 0.029, 0.019). A low level of the bone formation marker PINP was observed in the 20 M group, and a high level of the bone resorption marker TRAP5b was observed in the 10 M and 20 M groups. Micro-computed tomography (µ-CT) results showed that older rats had significantly higher bone formation than younger rats, with lower serum levels of PINP and higher levels of TRAP5b. The local environment of the old rat bone defects, surrounded by thickened bone, may have affected the results of our study. In conclusion, old rats showed greater new bone regeneration and healing capacity for round mandibular bone defects. This result was related to the fact that the bone defects in the 20 M rat group provided more favorable conditions for new bone regeneration.

## 1. Introduction

The restoration of maxillofacial bone defects remains a complex and challenging task for oral and maxillofacial surgeons. Various causes of bony defects in the jaw include odontogenic tumors and cysts, osteomyelitis, jaw osteonecrosis, traumatic injuries, and malignant lesions [1]. The influence of aging on bone healing or repair has been well documented in femur and tibia fracture areas, which represent reduced bone-healing capacity with aging [2,3,4]. In a rat model of facial bone fractures, mandibular condyle fracture healing was delayed by aging [5]. In addition to bone healing, young rats appear to show better bone formation than older rats in the treatment of femoral segmental defects [6]. Although the effects of aging on bone repair or healing have been well documented in the femur, tibia, and mandibular condyle fracture models [6], the influence of aging on the spontaneous healing of maxillofacial bone defects has not been fully elucidated.

Bone remodeling is a lifelong process coordinated by bone resorption by osteoclasts, followed by new bone deposition and formation in the resorbed area by osteoblasts [7]. With increasing age, a negative balance in bone remodeling occurs, and osteoblast number and activity decline, contrary to osteoclast activity, which does not change significantly [8,9]. Bone resorption exceeds new bone formation and leads to the loss of overall bone mineral density (BMD). The healing capacity of skeletal bone also changes with aging [10]. It is caused by biological changes in the age-related decline of osteogenic cells, vascularity of the periosteum, cytokine levels, and gene expression [2,11,12]. Age-related impaired bone healing has been well documented in preclinical studies [2,3]; however, very little is known about the influence of age-related changes in bone metabolism on maxillofacial bone healing [8]. The role of age-related mandibular healing remains controversial.

Changes in the bone remodeling process can be assessed by examining bone turnover markers. Procollagen type I N-terminal propeptide (PINP) is secreted by osteoblasts and is used as a bone formation marker [13]. The serum level of cross-linked C-telopeptide of type I collagen (CTXI) was used as a bone resorption marker. However, the correlation between age and changes in bone turnover levels remains controversial [14]. Many factors influence the levels of bone turnover markers, including age, sex, growth hormones, and menopause [15,16]. In general, aging leads to a change in the remodeling balance of bone formation and resorption and a reduction in the ability to form new bone [17]. The effect of age on changes in bone turnover markers has been documented in some clinical human studies; there was a significant difference in several bone turnover markers among different age groups [14,18,19]. However, the correlation between the serum levels of bone turnover markers and the healing capacity of maxillofacial bone defects has not been reported.

In animal studies, the rat mandibular angle area has been used to create critically sized bone defects [20]. This zone does not have teeth or tunnels for nerve travel and provides adequate space to create round bone defects. This round-shaped defect model of the mandibular angle area has been used for various in vivo studies to evaluate the efficacy of biomaterials or stem cells in repairing mandible defects [20,21]. However, this area has not been used to evaluate the effect of aging on spontaneous healing of the jawbone.

The effects of aging on maxillofacial bone regeneration remain unclear. In this study, we created a round bony defect in the mandibular angle area in rats of different age groups (2, 10, and 20 months) and evaluated the amount of regenerated bone radiologically and histologically. We compared the serum levels of bone turnover markers in different age groups and investigated the correlation between these markers and new bone regeneration. This study aimed to evaluate the effect of aging on the regeneration of mandibular bone in a round bony defect.

## 2. Material and Methods

### 2.1. Animals and Experimental Design

This study was approved by the Institutional Animal Care and Use Committee of the Jeonbuk National University Hospital, Jeonju, Republic of Korea (JBUH-IACUC-2020-18). A total of 21 male Sprague–Dawley rats aged 2, 10, and 20 months (Samtako Biokorea, Osan, Republic of Korea) were used in this study. The rats were housed one per cage under specific pathogen-free conditions in a specialized animal facility. The animals were fed a standard rodent diet and provided water ad libitum. The animals were acclimated to the new environment for 14 days before surgical intervention. The rats were divided into three groups: 2 months (2 M, n = 8), 10 months (10 M, n = 7), and 20 months (20 M, n = 6). The same mandibular bone defect diameter was created on the left side of the mandibular angle. Bone regeneration in this defect was evaluated using micro-computed tomography (µ-CT) and histological examination and compared among the age groups.

### 2.2. Surgical Intervention

Surgery was performed under general anesthesia by intramuscular injection of a combination of zoletil 50 (15 mg/kg; Vibac, Carros, France) and rumpun (0.2 mL/kg; Bayer Korea, Seoul, Republic of Korea). The left mandibular angle was shaved and disinfected with povidone-iodine. Local anesthesia was administered in this area with a subdermal injection of 2% lidocaine and epinephrine (1:100,000). One surgeon performed all the surgical interventions. A horizontal dermal incision was made in the left submandibular area along the inferior border of the mandible. Blunt dissection was performed, and the masseter muscle was exposed. The mandibular angle was visualized after incision and subperiosteal dissection of the masseter muscle. A 4 mm diameter round bone defect was created on the left mandibular angle with a trephine bur and a micromotor (Surgic Pro Plus, NSK, Kanuma, Japan) at approximately 2000 rpm under constant normal saline irrigation. After drilling, the residual bone chip was removed, and a defect was created. The muscles and skin were closed using a 3-0 Vicryl (ETHICON, Sommerville, NJ, USA). Gentamycin (1 mg/kg; Kookje, Seoul, Republic of Korea) and pyrin (0.5 mL/kg; Green Cross Veterinary Products, Seoul, Republic of Korea) were injected intramuscularly thrice daily for three days. Blood samples were obtained before sacrifice and 10 weeks after surgery. All rats were sacrificed, and mandibular specimens were obtained for further analysis.

### 2.3. µ-CT Analysis

Rat mandibular specimens were fixed in 10% formalin. All samples were analyzed using μ-CT at the Center for University-wide Research Facilities at Jeonbuk National University (Jeonju-si, Republic of Korea). The samples were taken using a SkyScan 1076 (Bruker, Kontich, Belgium) with a pixel size of 35 µm. The CT scanner was set to 100 kV voltage for the X-ray tube, 100 μA current for the X-ray source, and 190 ms exposure time. The detector and X-ray source were rotated by 0.6° in steps of 360°. Scanned images were reconstructed using NRecon software (Bruker, Ettlingen, Germany). The region of interest (ROI) for each sample was a round mandibular defect with a cylindrical shape. The diameter of the ROI was 4.0 mm, which covered the surgically created bone defects, and the height of the ROI was set considering the thickness of the bone defect. The total volume (TV), bone volume (BV), bone mineral density (BMD), trabecular thickness (TbTh), and trabecular space (TbSp) of the ROI of the regenerated bone in the defect were evaluated. Regenerated bones were reconstructed using three-dimensional images.

### 2.4. Serum Biochemical Analysis

Blood samples were obtained for serum analysis from the left jugular vein immediately before surgery at baseline (T0), 5 weeks after surgery (T1), and 10 weeks after surgery before sacrifice (T2). Blood samples were immediately delivered to the laboratory, and the serum was separated by centrifugation at 13,000 rpm for 10 min. All collected serum samples were stored in a deep freezer at −70 °C until analysis. Serum levels were analyzed by enzyme-linked immunosorbent assay (ELISA) using alkaline phosphatase (ALP, Cat. No. MBS7726999; MyBioSource, San Diego, CA, USA), PINP (cat. No. MBS7726503; MyBioSource), CTXI (cat. No. MBS7727020; MyBioSource), and tartrate-resistant acid phosphatase 5B (TRAP5b, Cat. No. MBS9901647; MyBioSource) according to the manufacturer’s instructions.

### 2.5. Histological Examination

Mandibular samples were decalcified in 10% EDTA for four weeks and dehydrated in ethyl alcohol and xylene. Samples were sectioned through the midline in the sagittal plane of the round mandibular defect and embedded in paraffin blocks. Paraffin blocks were sectioned and stained with hematoxylin and eosin. The sections show sagittal images of the mandibular defect and newly generated bone. Stained tissue slides were examined using an Olympus BX51 microscope (Olympus, Tokyo, Japan). Images of selected sections were acquired using a digital camera (DP-73; Olympus, Tokyo, Japan).

### 2.6. Immunohistochemistry

ALP and TRAP expression were evaluated by immunohistochemistry (IHC) of histological sections. Anti-ALP (M190; Takara) and anti-TRAP (sc-28204; Santa Cruz, CA, USA) were used as primary antibodies. The Dako REAL EnVision Detection System (Dako, Glostrup, Denmark) was used for immunohistochemical staining, according to the manufacturer’s instructions. Counterstaining was performed using Mayer’s hematoxylin (Sigma-Aldrich, St. Louis, MO, USA). The stained tissue slides were examined using an Olympus BX51 microscope (Olympus, Tokyo, Japan), and images were acquired using a digital camera (DP-73; Olympus, Tokyo, Japan).

### 2.7. Statistical Analysis

Variables of μ-CT and serum analysis were compared in three independent groups using one-way analysis of variance (ANOVA; Version 23, SPSS, Chicago, IL, USA). Dunnett’s T3 method was used for post hoc testing. The correlation between the parameters of μ-CT and bone formation markers was assessed using Pearson’s correlation coefficient. Differences were considered statistically significant when *p*-values were less than 0.05.

## 3. Results

### 3.1. Analysis of BV/TV, BMD, TbTh, and TbSp Using µ-CT

The BV/TV, BMD, TbTh, and TbSp values of each group were analyzed using μ-CT (Figure 1). The average BV/TV of the 2 M, 10 M, and 20 M groups were 21.12 ± 14.03, 33.09 ± 22.20, and 43.71 ± 13.23%, respectively (Figure 1A). The BV/TV of the 20 M group was significantly higher than that of the 2 M group (*p* = 0.029). The average BMDs of the 2 M, 10 M, and 20 M groups were 119.85 ± 118.59, 230.84 ± 193.67, and 326.88 ± 113.20 mg/cc, respectively (Figure 1B). The BMD of the 20 M group was significantly higher than that of the 2 M group (*p* = 0.019). The average TbTh of the 2 M, 10 M, and 20 M groups were 0.32 ± 0.05, 0.35 ± 0.07, and 0.38 ± 0.10 mm, respectively (Figure 1C). The average TbTh of the 20 M group was higher than that of the other groups. However, no significant differences were observed between the groups (*p* = 0.332). The average TbSp did not differ significantly among the 2 M, 10 M, and 20 M groups (0.48 ± 0.03, 0.48 ± 0.11, and 0.49 ± 0.12 mm, respectively, *p* = 0.941) (Figure 1D). There was a significant difference in the TV among the three groups (*p* < 0.001) (Figure 1E).

The sagittal images of the mandible bone defect of the μ-CT in each group are presented in Figure 2. The round bony defect was created, and the newly formed bone was observed at each end of the defect in the three groups. Especially, remarkable bony calcification was observed in the bone defects of the 10 M and 20 M groups. In the 10 M group, the newly formed bone was connected to each of the end edges of the defect, making a bony bridge across the defect (Figure 2B). And thick and calcified new bone was observed at the end edge in the defect of the 20 M group (Figure 2C).

Three-dimensional images of the mandibular specimen were reconstructed from the specimen μ-CT images (Figure 3). A round defect in the mandibular angle area was occupied by new bone, and new bone regeneration was observed. A round defect trace was observed, and new bone was generated from the surrounding defect. Shallow and thin new bones were observed, which were more prominent in the older age groups. The space not occupied by new bone was larger in the 2 M group than in the other groups (Figure 3A). In contrast, the bone defect in the 20 M group was almost completely occupied by new bone, and only a small round gap remained (Figure 3C).

### 3.2. Serum Levels of ALP, PINP, CTXI, and TRAP5b

The mean serum ALP levels in the 20 M group were higher than those in the 2 M and 10 M groups at T0, T1, and T2; however, the differences were not statistically significant (Figure 4A). The serum levels of PINP in the 2 M group were significantly higher than those in the 10 M (*p* = 0.026) and 20 M (*p* = 0.002) groups at T0 and the 20 M group (*p* = 0.006) at T2. Serum PINP levels in the 10 M group were significantly higher than those in the 20 M group (*p* = 0.020) at T2 (Figure 4B). There were no significant differences in the serum CTXI levels among the three groups at any time point. Serum TRAP5b levels in the 10 M group were significantly higher than those in the 2 M group at T0, T1, and T2 (*p* = 0.039, 0.016, and 0.024, respectively). TRAP5b levels in the 2 M group were lower than those in the 10 M and 20 M groups at all time points (Figure 4D).

There was a significant positive correlation between TRAP5b levels at T1 and BV/TV (r = 0.457, *p* = 0.037) and BMD (r = 0.475, *p* = 0.029). However, there was a significant negative correlation between PINP at T1 and TbSp (r = −0.511, *p* = 0.018) (Table 1).

### 3.3. Histological Examination of the New Bone Regeneration

The cross-sectional histological images of each group are shown in Figure 5. New bone regeneration occurred at both edges of the bone defects in all groups. In the 2 M group, part of the new bone was generated at the end edge of the bone defect. The defect was not fully filled with newly generated bone, and masticatory muscle tissue occupied the empty defect (Figure 5A). Favorable bone regeneration was observed in the 10 M and 20 M groups. In the 10 M group, remarkable bone regeneration was observed in the bone defect, and a bone bridge formed over both end edges of the defect (Figure 5B). Additionally, thick and mature bone regeneration was observed in the 20 M group (Figure 5C).

### 3.4. Analysis of ALP and TRAP Levels via IHC

The expression levels of ALP and TRAP are shown in Figure 6. New bone regeneration was observed in the bone defects of all groups, and the degree of new bone regeneration varied among groups. ALP expression was observed around osteogenic cells in the new bone matrix. In the histological images, the degree of ALP expression was not significantly different among the three groups (Figure 6A–C). There was no specific expression of TRAP in newly formed bone in any group (Figure 6E,F).

## 4. Discussion

Studies investigating the effects of aging on the healing capacity of maxillofacial bone defects are lacking. Generally, the healing capacity of a skeleton decreases with age [10]. In this study, we created a round bony defect in the mandibular angle area and investigated the effect of aging on new bone regeneration in rats of different ages (2-, 10-, and 20-month-old) using μ-CT analysis and histological examination. Ten weeks after creating the bony defect, the amount of regenerated bone on the defect was evaluated using the μ-CT data of BV/TV, BMD, TbTh, and TbSp. Blood samples were collected before surgery and 5 and 10 weeks after surgery, and the serum levels of bone turnover markers, including ALP, PINP, CTXI, and TRAP5b, were investigated to evaluate the correlation between bone turnover markers and new bone regeneration. We hypothesized that bone regeneration would be better in young rats than in older rats. Contrary to our hypothesis, the results of our study showed increased bone regeneration in the old rat group and delayed bone healing in an age-dependent manner in the mandibles of young rats.

The average BV/TV of the 2 M, 10 M, and 20 M groups were 21.12 ± 14.03, 33.09 ± 22.20, and 43.71 ± 13.23%, respectively (Figure 1A). There was a significant difference between the 2 M and 20 M groups (*p* = 0.029). The average BMD of the 2 M, 10 M, and 20 M groups were 119.85 ± 118.59, 230.84 ± 193.67, and 326.88 ± 113.20 mg/cc, respectively (Figure 1B). A significant difference between the 2 M and 20 M groups (*p* = 0.019) was observed. These results showed that new bone regeneration and the amount of new bone were age-dependent. The 20 M group had significantly higher bone regeneration than the 2 M group. These results were confirmed via histological examination. Histologically, the bone defect in the 2 M group was mostly occupied by the masticatory muscle tissue, and slight new bone regeneration was observed at the end of the bone defect margin. In contrast to the 2 M group, the older rats showed more new bone regeneration. The bone defect in the 10 M group was covered by newly formed bone, which was formed from both ends of the bone defect and connected (Figure 5B). In the 20 M group, favorable new bone regeneration that occupied the bone defects was observed. Mature and immature bones and fibrous tissues were regenerated in the bone defects of the 20 M group (Figure 5C).

There were differences in new bone formation among the groups; all groups showed some degree of new bone formation in the bone defects. ALP and TRAP expression in new bone tissues was investigated by IHC. ALP is an enzyme found on the surface of osteoblasts that plays an important role in hard tissue formation [22]. ALP expression was observed in the newly formed bone in all the groups, especially around the osteogenic cells inside the new bone matrix. Histologically, the degree of expression was not significantly different among the groups (Figure 6A–C). TRAP is highly expressed in osteoclasts, activated macrophages, and dendritic cells [23]. TRAP expression in newly formed bone was not specific to any group (Figure 6D–F).

The healing capacity of the maxillofacial bone has not been thoroughly investigated. Osteoblastic activity of the skeleton generally decreases with age. Age-related changes in BMD decrease in the jaw, similar to other sites of the skeleton [24]. The response to several adjuvant bone healing treatments also differs depending on age. When bone morphogenetic protein 2 is delivered into rat femoral defects, young rats show increased bone formation with increasing doses of the protein [6]. The biomineralized scaffold was grafted onto the cranial defects of mice, and delayed bone formation and a decreased quantity of bone tissue were observed in older mice [25]. When mechanical and electronic stimulations were applied to rats to induce osteogenic potential, greater bone quantity and quality increases were observed in young adult rats [26]. In young animals, mechanical stimulation induces an increase in bone formation that diminishes with increasing age [17]. Therefore, based on previous knowledge, we hypothesized that new bone regeneration and healing capacity in the mandibular bone defect model would be greater in young rats than in old rats. However, in contrast to previous reports, our study showed that new bone regeneration was more prominent in older rats.

The results of the serum level analysis showed that the levels of the bone formation marker PINP were significantly higher in the 2 M group than in the 10 M (*p* = 0.026) and 20 M (*p* = 0.002) groups at T0 and the 20 M group (*p* = 0.006) at T2. PINP levels in the 10 M group were significantly higher than those in the 20 M group (*p* = 0.020) at T2. However, ALP levels were not significantly different among the three groups. The expression of the bone resorption marker TRAP5b was significantly higher in the 10 M group than in the 2 M group at T0, T1, and T2 (*p* = 0.039, 0.016, and 0.024, respectively). TRAP5b levels in the 2 M group were lower than those in the 10 M and 20 M groups (Figure 4D). Bone turnover and metabolic marker levels were associated with age. The levels of PINP and beta-C-terminal telopeptide (β-CTx) in the serum are highly correlated with age, and patients under 20 years of age show significantly higher levels of PINP and β-CTx compared with older individuals [19]. In another study, the level of PINP was highest in the 20–29-year-old group and continuously decreased with increasing age [14,16]. A similar trend was observed in the present study. The serum PINP level was highest in the 2 M group and gradually decreased in the 10 M and 20 M groups.

Although higher levels of the bone formation marker PINP and lower levels of the bone resorption marker TRAP5b were observed in the 2 M group, the BV/TV and BMD in the 2 M group were significantly lower than those in the other groups (*p* = 0.029 and 0.019, respectively). Young rats with high levels of bone formation markers showed less new bone regeneration in the mandibular angle defect. This result was also observed in the correlation between bone formation marker levels and µ-CT parameters. PINP expression at T1 was significantly negatively correlated with TbSp (r = −0.511, *p* = 0.018). TRAP5b expression at T1 was significantly and positively correlated with BV/TV (r = 0.457, *p* = 0.037) and BMD (r = 0.475, *p* = 0.029) (Table 1). In a previous study, serum levels of the bone formation markers osteocalcin and PINP were higher in 5-month-old rats than in 20-month-old rats [26]. However, the relationship between differences in bone turnover markers in rats of different ages and bone regeneration-healing capacity has not been elucidated.

There was more new bone regeneration in the bone defect of the 20 M group than in the other groups. This was because new bone formation was affected not only by bone turnover markers but also by the environment of the bony defects. We attempted to create the same volume of round bone defects among the three groups to evaluate the effect of aging on the bone-healing capacity of the mandible. The 4-mm diameter round bone defect was made in the mandibular angle area using the same-sized trephine bur to set the same experimental conditions, except for age. However, the growth of the mandible differed among the three groups; older rats had larger mandibles with greater size and thickness. The thickness of the round bone defect in the old rats was greater than that in the young rats; therefore, the volume of the defect was greater (Figure 1E). The round bony defect in the old group was surrounded by thicker and greater bone than that in the young group, which was surrounded by very thin and small bone with a narrow marrow space. There was a significant difference in the total volume of the bone defects among the three groups (*p* = 0.000). The diameter of the defect in the three groups was 4 mm, indicating that the thickness of the bone defect was greater in old rats. This difference in the bone defect environment among the three groups might also have influenced the significant differences in BV/TV and BMD. Old rats have a bone defect surrounded by more thickened bone than young rats, which have a thin and narrow bone height, and this would be more beneficial for new bone regeneration and healing inside the defect.

The influence of systemic factors, such as sex, hormones, age, and metabolic disease, on the maxillofacial bone has not been established. The change in jaw bone density related to sex and age showed a pattern similar to that of other skeletons [24]. Older women have a higher tendency for bone mineral loss in the jaw than older men. Systemic osteoporosis is associated with loss of jaw BMD [24]. Although systemic disease is known to be a major factor in maintaining bone volume and architecture, local factors of functional and mechanical loads also play an important role in maintaining bone density [27]. Reducing mechanical loading on the mandibular condyle negatively influences bone density, whereas increasing age does not affect bone density [27]. In our study, the older rat group showed lower serum levels of PINP and higher levels of TRAP5b than the younger rats but showed a more favorable tendency for new bone regeneration and healing in the bone defect with higher BV/TV and BMD. Although the 20 M group showed decreased osteoblastic activity, contrary to the increase in osteoclast function under systemic conditions, the local environment of the 20 M group around the round bone defect in the mandibular angle area may provide favorable conditions for new bone growth.

In the present study, we investigated the effects of aging on new bone regeneration in rats with round mandibular bone defects. μ-CT showed that the older rat group had significantly higher bone formation than the younger rat group, with higher BV/TV and BMD. Although older rats had lower serum levels of PINP and higher levels of TRAP5b than younger rats, new bone formation was histologically more favorable in the former than in the latter. The local environment of bone defects in older rats was surrounded by thickened bones, which may have affected the results of our study. In conclusion, the older rat group showed greater new bone regeneration and healing capacity in the mandibular round bone defect, despite having a weak systemic condition with less osteoblastic activity. This result was related to the fact that the bone defects in the 20 M rat group provided more favorable conditions for new bone regeneration.

## Figures and Tables

**Figure 1 biomimetics-09-00466-f001:**
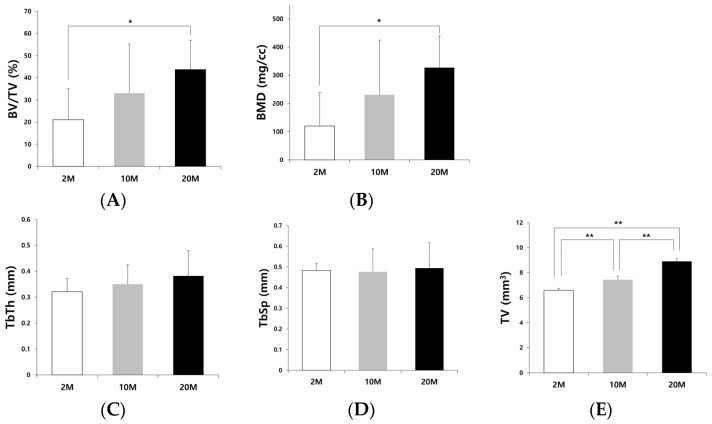
Micro-computed tomography analysis. (**A**) Bone volume/total volume (BV/TV), (**B**) bone mineral density (BMD), (**C**) trabecular thickness (TbTh), (**D**) trabecular space (TbSp), and (**E**) total volume (TV) of 2 M, 10 M, and 20 M groups. There was a significant difference in the BV, BMD, and TV among the three groups (* *p* < 0.05, ** *p* < 0.001).

**Figure 2 biomimetics-09-00466-f002:**
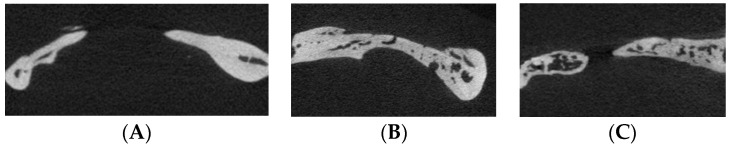
The sagittal images of the micro-computed tomography of the mandible bone defect of (**A**) 2 M, (**B**) 10 M, and (**C**) 20 M groups. The new bone regeneration was observed from the edge of the bone defect.

**Figure 3 biomimetics-09-00466-f003:**
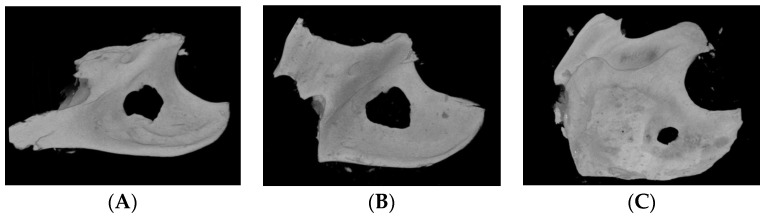
Three-dimensional (3D) reconstruction image of micro-computed tomography (μ-CT) of the mandible bone of (**A**) 2 M, (**B**) 10 M, and (**C**) 20 M groups. The new bone regeneration was more prominent in the 20 M group.

**Figure 4 biomimetics-09-00466-f004:**
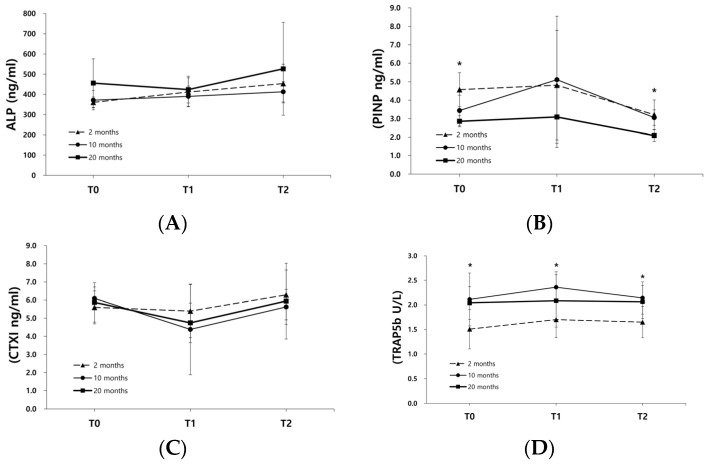
Serum levels of (**A**) ALP, (**B**) PINP, (**C**) CTXI, and (**D**) TRAP5b. PINP levels in the 2 M group were significantly higher than those in the 10 M and 20 M groups at T0 (*p* = 0.026 and 0.002, respectively) and the 20 M group at T2 (*p* = 0.006). PINP levels in the 10 M group were significantly higher than those in the 20 M group (*p* = 0.020) at T2. The serum TRAP5b levels in the 10 M group were significantly higher than those in the 2 M group at T0, T1, and T2 (*p* = 0.039, 0.016, and 0.024, respectively) (* *p* < 0.05).

**Figure 5 biomimetics-09-00466-f005:**
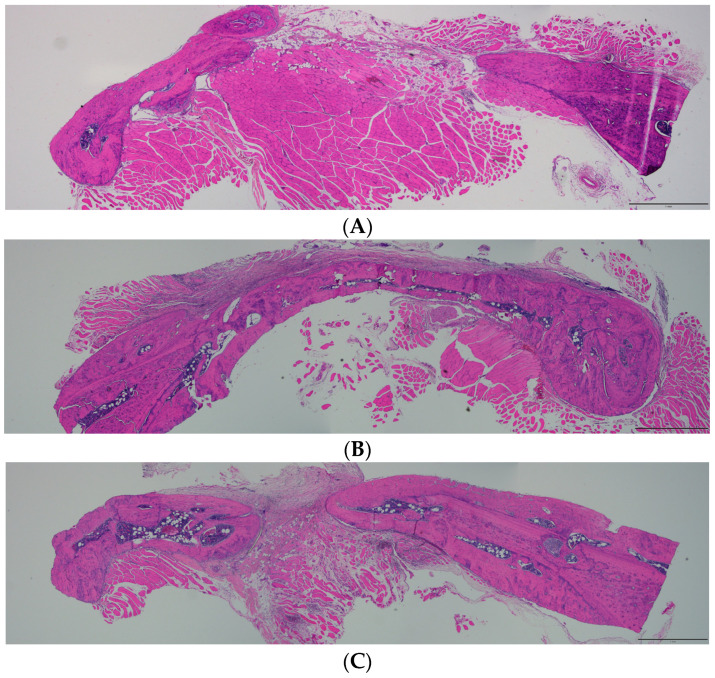
Histological images (hematoxylin and eosin staining) of each group. (**A**) 2 M, (**B**) 10 M, and (**C**) 20 M groups. A part of the new bone was formed at the end edge of the bone defect in the 2 M group. A remarkable bone regeneration was formed and covered the bone defect in the 10 M group. The thick and mature bone regeneration was observed in the 20 M group (original magnification 4×, scale bar = 1 mm).

**Figure 6 biomimetics-09-00466-f006:**
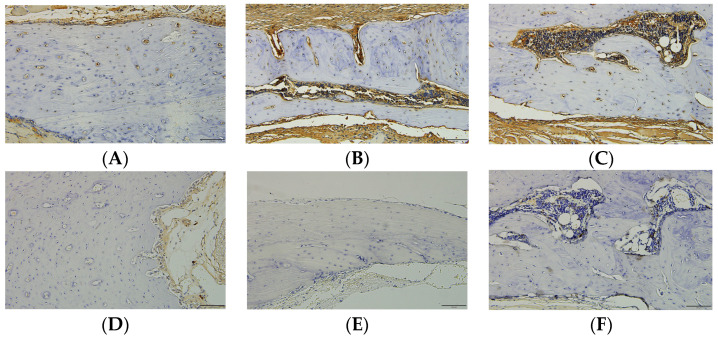
Immunohistochemical staining of ALP (**A**–**C**) and TRAP (**D**–**F**) in each group. (**A**,**D**) 2 M, (**B**,**E**) 10 M, and (**C**,**F**) 20 M group. ALP expression was observed in the osteogenic cells in the new bone matrix. TRAP expression was not specific in all groups (magnification 100×, scale bar = 100 µm).

**Table 1 biomimetics-09-00466-t001:** Correlation of the parameters of μ-CT and bone turnover marker.

	PINP (T1)		TRAP5b (T1)	
	r	*p* Value	r	*p* Value
BV/TV	0.358	0.111	0.457	0.037 *
BMD	0.346	0.125	0.475	0.029 *
TbSp	−0.511	0.018 *	−0.372	0.096

BV/TV, bone volume/total volume; BMD, bone mineral density; TbSp, trabecular bone space; PINP, procollagen type I N-terminal propeptide; TRAP5b, tartarate resistant acid phosphatase 5B; r, Pearson correlation coefficient; *, *p* < 0.05.

## Data Availability

Data is contained within the article.

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
