# Peer review of "Effects of Aging on New Bone Regeneration in a Mandibular Bone Defect in a Rat Model"

_biomimetics, 2024, doi:10.3390/biomimetics9080466_

Round 1

Reviewer 1 Report

Comments and Suggestions for Authors

Following points should be addressed in the manuscript before the manuscript can considered for publication:

1) Authors need to update the abstract. The abstract should highlight the problem statement and why the manuscript is needed to solve that problem. The current abstract is very data-heavy and might be difficult to comprehend for readers not aware of the field.

2) In the experimental design, the authors need to include more details on how the micro-CT was performed and how the TV, BV, BMD, TbTh, and TbSp were calculated and estimated using the micro-CT images. If multiple images were taken per sample for analysis, those details should be included as well.

3) In the current manuscript, it is unclear what is the significance of the TV, BV, BMD, TbTh, and TbSp measurements and what do these measurements tell. Authors are recommended to include more details before discussing the results. 

4) Figure 1E is not discussed in the results section. Is there a reason why the discussion of Figure 1E was omitted? Please discuss it in the manuscript.

5) For the micro-CT, please include the raw images in the SI for broader audience understanding. Authors are also encouraged to include a representative image for each condition in the main text alongside the graphs discussing the parameters assessed from micro-CT images.

6) In general, for the results section, expand more on the results section and specifically highlight the importance of each study/assay the authors are performing to better connect and understand the results from individual figures with rest of the manuscript.

Author Response

1) Authors need to update the abstract. The abstract should highlight the problem statement and why the manuscript is needed to solve that problem. The current abstract is very data-heavy and might be difficult to comprehend for readers not aware of the field.

-> Thanks for your kind advice. The abstract our manuscript was too long and complicatedly described according to your advice. We revised the abstract that can be easily understood the result and conclusion for the reviewers.

2) In the experimental design, the authors need to include more details on how the micro-CT was performed and how the TV, BV, BMD, TbTh, and TbSp were calculated and estimated using the micro-CT images. If multiple images were taken per sample for analysis, those details should be included as well.

-> Thanks for your kind advice. The micro CT machine and setting for the CT taking were described well and in detail. But, as your advice, the explanation of the setting of the ROI was simply described. We add more detail procedure of the setting of the ROI in the material and method 2.3.

3) In the current manuscript, it is unclear what is the significance of the TV, BV, BMD, TbTh, and TbSp measurements and what do these measurements tell. Authors are recommended to include more details before discussing the results.

-> Thanks for your kind advice. The explanation of the each parameter was described in the material and method 2.3. And the meaning of the those parameter result and significance among three groups was described in second paragraph in the discussion.  

4) Figure 1E is not discussed in the results section. Is there a reason why the discussion of Figure 1E was omitted? Please discuss it in the manuscript.

-> Thanks for your kind advice. We add the result of the TV among these groups in the result section 3.1.

5) For the micro-CT, please include the raw images in the SI for broader audience understanding. Authors are also encouraged to include a representative image for each condition in the main text alongside the graphs discussing the parameters assessed from micro-CT images.

-> Thanks for your kind advice. We though that the main results parameter of the micro-CT were BV/TV and BMD. We added the sagittal view of the micro CT image in each group that the audience can be easily understood the result of this parameters.

6) In general, for the results section, expand more on the results section and specifically highlight the importance of each study/assay the authors are performing to better connect and understand the results from individual figures with rest of the manuscript.

-> Thanks for your kind advice. The main result of this study was that the older rats had significantly higher bone formation than younger rats, with lower serum levels of PINP and higher levels of TRAP5b. The higher bone formation was well described in the micro CT result and the histology image was supported the result of the micro CT. Although this contrary result of the micro CT and serum level analysis was hard to explain and elucidate, but we thought that this result was more related the local environment of the bone defect rather than general condition of the bone metabolism marker. Those result was discussed in the 7th paragraph in the discussion. 

Reviewer 2 Report

Comments and Suggestions for Authors

The article is related to a study on the effects of aging on bone regeneration in a mandibular bone defect in a rat model.

The study is original and interesting, anyway the authors should give some clarifications:

Materials and methods:

1) The total number of rats is equal to 21 but the groups are not composed by the same number of rats (2M, n=8; 10M, n=7 and 20M, n=6). Can the authors justify these differences and how do the authors apply the statistical analysis? The authors should add more details related to the statistical approach.

2) The pixel size of the micro-CT scanner is equal to 35 micron. Is the dimension sufficient? there are several research groups that analyse scansions from synchrotron  with resolution equal to about 1 micron (see the articles of Prof. Ralph Muller (ETH, Zurich) and  prof. Laura Vergani (Politecnico di Milano). Can the authors consider these different approaches?

Results:

1) The authors should add a reference dimension to the 3D reconstruction  and to the histological images  in order to allow a better comprehension of the phenomena.

Besides, at pag.3 row 106 the reference at fig 1 is not correct.

Author Response

Materials and methods:

  • The total number of rats is equal to 21 but the groups are not composed by the same number of rats (2M, n=8; 10M, n=7 and 20M, n=6). Can the authors justify these differences and how do the authors apply the statistical analysis? The authors should add more details related to the statistical approach.

-> Thanks for your kind advice. We performed the animal experiment and the number of the animal was differed in each group. The old rats that were aged 10 and 20 months was hard to be supplied because of that the difficult of the growing. So in our state, it was best to be supplied those number of the 10 and 20 months rats. There was some difference in the number of the each rats, that was not big enough to be impossible the statistical analysis. In those objects, we could statistically analyze the a number of parameters from our experiments.

2) The pixel size of the micro-CT scanner is equal to 35 micron. Is the dimension sufficient? there are several research groups that analyse scansions from synchrotron with resolution equal to about 1 micron (see the articles of Prof. Ralph Muller (ETH, Zurich) and  prof. Laura Vergani (Politecnico di Milano). Can the authors consider these different approaches?

-> Thanks for your kind advice. As your advice, more small and detail pixel size would be more accurate for the evaluation. But, to the setting of the pixel size, we thought that the several things should considered such as sample size, cost, shooting time, and etc. The anyone can think that the 35 micron quite big and it would be inaccurate, but the experiment of our study was used rat and the defect diameter of our sample was more than 4 mm and we thought the pixel size of the 35 micron would be enough to analyze the bone regeneration among groups. There are several studies that micro CT pixel size was set from 10-30 in the rat model. (DOI: 10.1096/fj.201902779R, doi: 10.7150/ijms.32590, Journal of Oral Science, Vol. 61, No. 4, 534-538, 2019, DOI: 10.1002/jbm4.10068)

Results:

1) The authors should add a reference dimension to the 3D reconstruction and to the histological images in order to allow a better comprehension of the phenomena.

-> Thanks for your kind advice. We presented the scale bar on the left lower in the figure of histology of H&E and IHC. But the scale bar in IHC was so thin, according to your advice it is hard to identified.

The Fig.2 of 3D reconstruction image was captured Print Screen in the NRecon software. We trided to present the reference dimension such as scale bar in this program but this software does not provide this function. I really sorry to say it is hard to add the reference dimension in the 3D reconstruction image.

Besides, at pag.3 row 106 the reference at fig 1 is not correct.

-> Thanks for your kind advice. We delete the reference at fig 1.

Round 2

Reviewer 1 Report

Comments and Suggestions for Authors

Authors have addressed my comments. Manuscript can be accepted for publication.